# New Diagnostic and Prognostic Models for the Development of Alcoholic Cirrhosis Based on Genetic Predisposition and Alcohol History

**DOI:** 10.3390/biomedicines11082132

**Published:** 2023-07-28

**Authors:** Monica Mischitelli, Alessandra Spagnoli, Aurelio Abbatecola, Claudia Codazzo, Marta Giacomelli, Simona Parisse, Rosellina Margherita Mancina, Claudia Rotondo, Fabio Attilia, Stefano Ginanni Corradini, Flaminia Ferri

**Affiliations:** 1Department of Translational and Precision Medicine, “Sapienza” University of Rome, Viale dell’Università 37, 00185 Rome, Italy; monica.mischitelli@uniroma1.it (M.M.); aurelioabbatecola@yahoo.it (A.A.); giacomelli.marta95@gmail.com (M.G.); simona.parisse@uniroma1.it (S.P.); flaminia.ferri@uniroma1.it (F.F.); 2Section of BioMedical Statistics, Department of Public Health and Infectious Disease, Sapienza University, 00185 Rome, Italy; alessandra.spagnoli@uniroma1.it; 3Department of Mental Health ASL RM1, UOSD CRARL, Sapienza University, 00186 Rome, Italy; claudia.codazzo@aslroma1.it (C.C.); claudia.rotondo@aslroma1.it (C.R.); fabio.attilia@aslroma1.it (F.A.); 4Wallenberg Laboratory, Department of Molecular and Clinical Medicine, Institute of Medicine, University of Gothenburg, 40530 Gothenburg, Sweden; rosellina.mancina@wlab.gu.se

**Keywords:** alcoholic liver disease, cirrhosis diagnosis, cirrhosis prediction, genetics, HSD17B13, PNPLA3

## Abstract

Liver cirrhosis development is a multifactorial process resulting from a combination of environmental and genetic factors. The aim of the study was to develop accurate non-invasive diagnostic and prognostic models for alcoholic cirrhosis. Consecutive subjects with at-risk alcohol intake were retrospectively enrolled (110 cirrhotic patients and 411 non-cirrhotics). At enrollment, the data about lifetime drinking history were collected and all patients were tested for Patatin-like phospholipase domain-containing protein 3 (PNPLA3) rs738409, Transmembrane 6 Superfamily 2 (TM6SF2) rs58542926, and hydroxysteroid 17-beta dehydrogenase 13 (HSD17B13) rs72613567 variants. In cross-sectional analyses, models for the diagnosis of cirrhosis were developed using multivariate logistic regression. A predictive score for cirrhosis development over 24 years was built by evaluating time-dependent AUC curves. The best diagnostic accuracy was demonstrated by the model, which also includes daily alcohol consumption, duration of hazardous alcohol use, and genetic variants, with AUCs of 0.951 (95% CI 0.925–0.977) and 0.887 (95% CI 0.925–0.977) for cirrhosis and compensated cirrhosis, respectively. The predictive model for future cirrhosis development (AUC of 0.836 95% CI: 0.769–0.904) accounted for age at onset of at-risk alcohol consumption and the number of PNPLA3 and HSD17B13 variant alleles. We have developed accurate genetic and alcohol consumption models for the diagnosis of alcoholic cirrhosis and the prediction of its future risk.

## 1. Introduction

In 2017, nearly 123 million individuals were suspected of having alcohol-related cirrhosis worldwide [1]. Mortality is twice as high in patients with alcohol-related cirrhosis as in those with cirrhosis from other causes [2]. In 2019, heavy alcohol consumption was associated with 25% and 42% of cirrhosis mortality and 19% and 35% of liver cancer mortality worldwide and in Europe, respectively [3]. In 2019, the global age-standardized death rate for alcohol-related cirrhosis and liver cancer was 4.5 and 1.1 deaths per 100,000 people, respectively [3].

Making a diagnosis of cirrhosis in subjects with heavy alcohol consumption is important from a prognostic and motivational point of view. In fact, in a population-based Danish hospital registry study of patients with predominantly alcoholic cirrhosis, survival 10 years after cirrhosis diagnosis was reported to be only 22% [4]. Survival also depends on the stage of the disease in which cirrhosis is diagnosed, with median values of 12 and 2 years, respectively, for subjects with compensated and decompensated cirrhosis [5]. These conclusions are based on a systematic review of 118 studies, of which 58% were prospective, 51% included consecutive patients, and 26% met the authors’ criteria for a good-quality study, which evaluated predictors of long-term survival (>6 months) in patients with cirrhosis [5]. Furthermore, the diagnosis of cirrhosis may motivate patients to stop alcohol intake in order to improve the prognosis and, in more advanced cases, to be able to access liver transplantation. 

The clinical diagnosis of liver cirrhosis in heavy alcohol users is based on the presence or history of complications of this disease, liver imaging, and blood tests of liver function and platelet counts [6]. Several easy-to-perform and low-cost non-invasive bioassays have also been developed to rule out advanced fibrosis or the presence of cirrhosis in alcoholics even when compensated, including the Fibrosis-4 Index (FIB-4) and the AST to Platelet Ratio Index (APRI) [7,8,9,10,11,12].

The first aim of the present study was to improve the diagnostic accuracy of non-invasive tests for alcoholic cirrhosis. To do this, we also considered the patient’s detailed history of alcohol consumption, genetic predisposition to alcohol-induced liver injury, and the presence of metabolic syndrome features. The amount of alcohol consumed by the patient over time is in fact associated with the development of cirrhosis [13]. Genetic variants have also been described, which, given the same alcohol consumption, could be associated with a different susceptibility to the development of alcoholic cirrhosis. Among them, PNPLA3, rs738409 C > G, p.I148M, transmembrane 6 superfamily 2 (TM6SF2), and rs58542926 (C/T) E167K variants have been associated with the development and progression of alcoholic liver disease [14,15]. In contrast, another gene variant, a splice variant named rs72613567 of the hydroxysteroid 17-beta dehydrogenase 13 (HSD17B13) gene, was found to be associated with a protective effect against disease progression [14]. Recent data suggest a clinical utility of polygenic risk scores for the diagnosis of advanced fibrosis or alcoholic cirrhosis, also taking into account patients’ metabolic variables [16,17]. In fact, the interaction of metabolic syndrome or its components with heavy alcohol consumption increases the risk of liver disease [18]. In addition, the genetic variants PNPLA3 rs738409, TM6SF2 rs58542926, and HSD17B13 rs72613567 are involved in lipid droplet metabolism in hepatocytes and have been associated with both alcoholic and nonalcoholic fatty liver disease [16,19,20].

In addition to improving the non-invasive diagnostic accuracy for alcoholic cirrhosis, it would be important, from the point of view of both prevention and future cost calculation strategies, to also have a predictive score of the development of alcoholic cirrhosis over time. Thus, the second objective of the present study was to develop a prognostic score of subsequent development of alcoholic cirrhosis. We previously demonstrated that age at the start of at-risk alcohol use and the PNPLA3 rs738409 variant were associated with future risk of developing cirrhosis [21]. In the present study, we developed a new model considering all three aforementioned genetic variants, adjusting for known variables at the time of onset of at-risk alcohol consumption, such as Body Mass Index (BMI) and gender [18,22]. 

We found that our diagnostic models of alcoholic cirrhosis all performed better than FIB-4 and APRI. Furthermore, as far as the diagnosis of compensated cirrhosis is concerned, our models, we developed different models taking in account alcohol consumption and genetics to liver fat accumulation achieving progressively greater accuracies.

Finally, the HSD17B13 rs72613567 variant, in addition to the PNPLA3 rs738409 variant, also allows good accuracy of the prognostic score for the future development of alcoholic cirrhosis to be achieved based on the age of onset of at-risk alcohol consumption.

## 2. Materials and Methods

### 2.1. Population and Study Design

A total of 2182 consecutive Caucasian patients followed in the liver disease outpatient services of the Translational and Precision Medicine, Policlinico Umberto I, Rome, Italy, between 2010 and 2021 were retrospectively analyzed. On the first visit, all patients underwent a detailed clinical examination and interview on their clinical history and lifestyle habits. At-risk alcohol consumption, defined as ≥3 and ≥2 alcohol units per day for men and women, respectively, for at least 5 years, was present in 1180 patients. One unit of alcohol was defined as 12 g of ethanol. In patients with at-risk alcohol use, a more thorough alcohol history was obtained by interview, performed by specialist experts in alcohol history, using the lifetime drinking history (LDH). In particular, the age reported at the beginning of at-risk alcohol consumption, the duration in years of at-risk alcohol consumption, and the average daily intake of alcohol, expressed as the number of alcoholic units per day, were obtained. The eventual presence of cirrhosis and the time of first diagnosis of cirrhosis were also reported, as previously described [23]. Body mass index (BMI) calculation based on dry weight was recorded. Patients were asked whether their body weight was stable (changes < 5 kg) compared to that at age 25 at the first visit for noncirrhotic patients and before the diagnosis of cirrhosis in cirrhotic patients. Subjects with a history of unstable BMI, a present or previous concurrent diagnosis of hepatitis B and/or C, autoimmune hepatitis, primary biliary cholangitis, primary sclerosing cholangitis, Wilson’s disease, and hemochromatosis were excluded from the study. We also excluded from the analyses subjects with incomplete LDH data, clinical/biochemical data that were too incomplete to obtain a diagnosis of cirrhosis and/or a definite age at diagnosis of cirrhosis, or with poor DNA quality. A total of 521 patients were included in the study, as described in the flowchart (Figure 1).

By analyzing the entire study population in a cross-sectional way, we wanted to develop predictive indices for the diagnosis of cirrhosis that also took into account the patients’ alcoholic history and their genetic predisposition to alcohol-induced liver damage. We then compared the accuracy of our predictive indices with that of other widely used indices, such as the APRI and the FIB-4. Since the utility of a diagnostic model for cirrhosis is limited when cirrhosis is advanced and therefore clearly diagnosable clinically, we repeated the cross-sectional analysis, limiting it, as far as cirrhotic patients were concerned, to those with compensated cirrhosis. Compensated cirrhosis was defined as cases without previous or current ascites, variceal hemorrhage, hepatic encephalopathy, or jaundice.

Finally, we aimed to develop a predictive risk model of future development of cirrhosis in heavy drinkers. To do this, we considered the variables inferable at the time of the onset of at-risk alcohol consumption, including genetic predisposition. Informed consent was obtained from all subjects involved in the study. The study was conducted in accordance with the Declaration of Helsinki and approval was obtained from the local ethical committee Cet—Comitato Etico Lazio Area 1 (protocol code 1913/18.11.2010, date of approval 18 November 2010). 

### 2.2. Polymorphism Screening

Ethylene-diamine-tetra acetic acid (EDTA) blood, obtained by venepuncture, was collected to recover DNA. 

DNA was obtained by Peripheral Blood Mononuclear Cells (PBMC) according to the manufacturer’s instructions (QIAmp^®^ DNA Blood Kit, QIAGEN S.p.A, Milan, Italy). Purified DNA was quantified by O.D. 260 and directly used for polymorphism screening.

The rs738409 (I148M, PNPLA3) and rs58542926 (E167K, TM6SF2) single-nucleotide polymorphisms (SNPs) were detected using specific TaqMan^®^ Predesigned SNP Genotyping Assays. Each assay includes two allele-specific TaqMan^®^ MGB probes containing distinct fluorescent dyes and a PCR primer pair to detect the specific target. A total of 1–20 ng purified genomic DNA was used per well (final concentration: 0.2 ng/μL).

The identification of the rs72613567 variant of the HSD17B13 gene was carried out using a custom Taqman assay designed in house and produced by Thermo Fisher Scientific Italy.

Genotyping data were confirmed through automatic sequencing. In particular, in order to generate an amplicon suitable for sequencing, Primer BLAST tool software (https://www.ncbi.nlm.nih.gov/tools/primer-blast/, accessed on 13 December 2019) was used to design primers surrounding the sequence of interest and generate a 677 bp amplicon.

The Basic Local Alignment Search Tool analyzed sequences acquired from the NCBI website, whereas alignments were performed with ClustalW2 at the EMBL-EBI website using default parameters.

For all variants, genotyping was performed in triplicate with 100% concordance between replicates.

### 2.3. Statistical Analyses

For continuous variables, the normality was assessed by the Shapiro–Wilk test. Since the data distribution for continuous variables was non-normal, data were reported as medians with interquartile ranges (IQR). For categorical variables, data were reported as frequencies and percentages. Baseline characteristics were compared by the Kruskal–Wallis test or Chi square test, as appropriate. In multivariable models, only independent variables that were not highly correlated were considered. In cross-sectional analyses for cirrhosis diagnosis, a multivariable logistic regression model was estimated to establish the influence of covariates: sex, BMI, platelet count (10^3^/µL), serum total bilirubin (mg/dL), ALT (U/L), and creatinine (mg/dL), as well as the presence of diabetes and/or dyslipidemia and/or arterial hypertension, on the outcome (Model 0). Model selection was performed by a stepwise procedure based on the Akaike Information Criterion. To establish the influence on the outcome of the history of alcohol, Model 1 was defined, including the following covariates in Model 0: daily alcohol consumption (unit) and duration of at-risk alcohol consumption (years). Model 2 included the Model 1 covariates and genetic components. The diagnostic accuracy of multivariable logistic regression models was assessed by the area under the curve (AUC) plotting receiver operating characteristic (ROC) curve that was designed to differentiate between patients with and without cirrhosis. The ROC curves obtained with the different models were compared with Delong’s test. The multivariable logistic regression models were validated by means of cross-validation methods using 10-fold cross-validation. 

In time-dependent longitudinal analyses for the risk of future cirrhosis development, time of progression to cirrhosis was defined as the time to onset of at-risk alcohol consumption to diagnosis of cirrhosis. Patients who did not have a cirrhosis diagnosis were censored at the date they were last known to be alive. The cirrhosis probabilities were estimated in each group using the non-parametric Kaplan–Meier method and displayed graphically. 

The group differences in cumulative incidence across genotypes were assessed by log–rank test. The multivariable Cox proportional hazard model was used to evaluate the effect of age at the start of at-risk alcohol consumption, PNPLA3, and HSD17B13 SNPs on the risk of developing cirrhosis after adjusting for sex and BMI. The proportional hazards assumption was verified using graphical methods; scaled Schoenfeld residuals and graphical checks proposed by Klein and Moeschberger were performed.

A score to predict alcoholic cirrhosis incidence at 24 years from the onset of at-risk alcohol consumption was defined as the linear combination of the predictors in the Cox regression model, where the weights were proportional to the corresponding rounded coefficients. The performance of the prognostic score was assessed using time-dependent receiver operating characteristic (ROC) curves.

The optimal cut-off value was chosen to maximize the sum of the sensitivity and specificity (Youden index).

Internal validation was obtained by means of 10,000 bootstrap replicates. All analyses were performed using R software (version 4.2.2, R Foundation for Statistical Computing, Vienna, Austria).

## 3. Results

### 3.1. Cross-Sectional Study in the Entire Study Population

To define a predictive index for the diagnosis of cirrhosis, we first analyzed the entire study population in a cross-sectional manner. As shown in Table 1, we compared 411 non-cirrhotic patients with heavy alcohol consumption and 110 patients with alcoholic cirrhosis. As expected, cirrhotic patients compared with non-cirrhotic patients had significantly higher INR, serum bilirubin, creatinine, and AST values. Cirrhotic patients compared with non-cirrhotic patients were significantly older, had a higher BMI, and were more frequently affected by diabetes and arterial hypertension but less frequently by dyslipidemia. Regarding the history of alcohol consumption, cirrhotic patients compared with non-cirrhotics had lower daily alcohol consumption, but the duration of at-risk alcohol consumption was longer. There was no difference between the two groups for age at onset of at-risk alcohol consumption. Regarding genetics, the group of cirrhotic patients, compared to that of non-cirrhotics, had a significantly higher frequency of the PNPLA3 rs738409 variant, particularly in homozygosity. The protective rs72613567 HSD17B13 variant (heterozygosity) was significantly less frequent in the cirrhotic patient group (Table 1). No difference was observed for the rs58542926 TM6SF2 variant.

Diagnostic models for the presence of cirrhosis in the whole study population were defined. Post hoc sensitivity analyses showed the linear relationship between any continuous independent variable and log odds. Moreover, the multicollinearity concern was addressed through the variance inflation factor, which was found to be lower than the critical value [24]. Model 0 was defined using only biochemical and clinical variables significantly associated with the diagnosis of cirrhosis. Model 1 was defined considering both biochemical and clinical variables, as well as alcohol history variables significantly associated with the diagnosis of cirrhosis. Finally, Model 2 was defined considering both the biochemical and clinical variables, those relating to the history of alcohol, and genetic ones significantly associated with the diagnosis of cirrhosis (Figure 2A–C).

To test the accuracy of cirrhosis diagnosis of our models, we compared them with that of FIB4 and APRI in the entire study population (Figure 3). As described in Table 2, all the proposed models obtained significantly better results than FIB4 and APRI. In particular, the AUROCs were 0.791 for APRI, 0.877 for FIB-4, 0.941 for Model 0, 0.949 for Model 1, and 0.951 for Model 2.

When applied to the cross-validation method considering 10 folds the AUROCs were 0.941 (95% CI: 0.913–0.968) for Model 0, 0.946 (95% CI: 0.921–0.971) for Model 1, and 0.949 (95% CI: 0.923–0.974) for Model 2.

### 3.2. Cross-Sectional Study Limited to Cirrhotic Patients with Compensated Cirrhosis

Since the clinical utility of a diagnostic model for cirrhosis is greatest when the cirrhosis is not advanced, we repeated the analysis considering only cirrhotic patients with compensated cirrhosis. For this analysis, we therefore compared the 411 non-cirrhotic patients with heavy alcohol consumption and 38 patients with alcoholic cirrhosis without any previous or present cirrhosis complication. In the univariate analysis (Table 1), the same differences between non-cirrhotic and cirrhotic patients were found as those we had found considering the entire study population except that there were no significant differences in BMI, daily alcohol consumption, or AST serum activity.

Diagnostic models for the presence of compensated cirrhosis were built (Figure 4).

Model 0 for compensated cirrhosis was constructed using only biochemical and clinical variables significantly associated with the diagnosis of cirrhosis. Model 1 considered both biochemical and clinical variables, as well as alcohol history variables significantly associated with the diagnosis of cirrhosis, and, finally, Model 2 included both biochemical and clinical variables, those related to alcohol history, and genetic variables significantly associated with the diagnosis of cirrhosis.

We then compared the diagnostic accuracy for cirrhosis of our models with that of FIB4 and APRI (Figure 5). As described in Table 3, all of the proposed models obtained significantly better results than FIB4 and APRI. In particular, the AUROCs were 0.655 for APRI, 0.737 for FIB-4, 0.835 for Model 0, 0.868 for Model 1, and 0.887 for Model 2.

When applied to the cross-validation method, considering 10 folds, the AUROCs were 0.837 (95% CI: 0.766–0.901) for Model 0, 0.866 (95% CI: 0.802–0.929) for Model 1, and 0.890 (95% CI: 0.831–0.950) for Model 2.

### 3.3. Longitudinal Study

Since the rs738409 PNPLA3 and rs72613567 HSD17B13 but not the rs58542926 TM6SF2 were significantly associated with the development of liver cirrhosis over time (Appendix A), we delineated a predictive model for cirrhosis development over time. This model considered the variables that can be inferred at the time of the onset of alcohol consumption.

The multivariable Cox regression model, adjusted for sex and BMI, shown that the development of cirrhosis was significantly associated with age at the start of at-risk alcohol consumption (HR = 1.10, 95% CI: 1.07–1.12, *p* < 0.001), PNPLA3 (HR = 2.36, 95% CI: 1.44–3.87, *p* < 0.001), and HSD17B13 variants (HR = 0.62, 95% CI: 0.40–0.96, *p* = 0.03). The AUROC for the predictive model of cirrhosis development at 24 years was 0.836 (95% CI: 0.755–0.918). The prognostic score was defined as (0.09 × age at onset of at-risk alcohol consumption) + (0.86 × number of PNPLA3 variant alleles) − 0.47 × number of HSD17B13 variant alleles). The threshold of 3.34 performed the prediction of the risk of alcoholic cirrhosis development, with a sensitivity 0.74 and a specificity of 0.75. The score was internally validated by the bootstrap sampling procedure, yielding an AUC of 0.777 (95% CI: 0.702–0.845).

## 4. Discussion

In our study, we retrospectively analyzed a cohort of heavy drinkers characterized in detail from the clinical hepatic point of view with regard to the metabolic syndrome, the history of alcoholic habits, and the genetic predisposition to alcoholic liver damage (Appendix A). We carefully subgrouped patients into non-cirrhotic and cirrhotic, excluding from the study those with unclear stages of liver disease. We then analyzed the data in a cross-sectional fashion to derive diagnostic models for cirrhosis and for compensated cirrhosis. Knowing the onset time of at-risk alcohol consumption and that of the eventual diagnosis of cirrhosis, we analyzed the data in longitudinal mode to develop a predictive model of the development of alcoholic cirrhosis over time.

The first result of our study, obtained with the cross-sectional analyses, is that we developed diagnostic models for alcoholic cirrhosis that are more accurate than FIB-4 and APRI, two widely used indirect markers of liver fibrosis. Our basic model (model 0) is based on a few commonly used laboratory tests, such as platelet count, serum bilirubin, creatinine, and ALT, associated with the presence or absence of diabetes, arterial hypertension, and dyslipidemia. By adding to the basic model daily alcohol consumption and the duration of at-risk alcohol consumption (model 1), as well as the presence or absence of predisposing genetic variants (model 2), we progressively increased the diagnostic accuracy of the diagnosis of cirrhosis in the entire study population, resulting in an AUC of 0.951. In a recent study, the AUCs for the diagnosis of cirrhosis of different genetic models with diabetes status ranged from 0.619 to 0.665 [16]. Our AUC values for the diagnosis of cirrhosis were higher for the greater number of clinical variables and the detailed alcoholic variables available. In our study, a progressive improvement in diagnostic accuracy was especially achieved for the diagnosis of compensated cirrhosis, where clinical diagnosis is more difficult. Indeed, whereas the AUC of FIB-4 was 0.737, it was 0.868 and 0.887 for Model 1 and Model 2, respectively. In non-specialized units where transient elastography is not available, blood tests recommended for diagnosing compensated alcoholic cirrhosis include, in addition to the platelet count and serum AST and ALT used to calculate FIB-4 and APRI, INR and bilirubin, albumin, and serum creatinine [10]. Our models save the costs of measuring INR and serum AST and albumin. Accurate alcohol history and the addition of genetic testing for PNPLA3 rs738409 and HSD17B13 rs72613567 variants allow our models to increase diagnostic accuracy for compensated alcoholic cirrhosis, with low additional costs. This is also true for Model 2, because the cost of genetic testing is progressively decreasing.

The second result of our study, obtained with the longitudinal analysis, is that we outlined a predictive model of the development of alcoholic cirrhosis in the 24 years following the onset of at-risk alcohol consumption. The model, adjusted for sex and BMI, is based on the patient’s age at the onset of at-risk alcohol use and on the PNPLA3 rs738409 and HSD17B13 rs72613567 variants. A score with a cutoff of 3.34 demonstrates a good sensitivity and specificity for predicting the development of cirrhosis in patient with heavy alcohol consumption. We previously developed a prognostic model of future development of alcoholic cirrhosis in males that took into account BMI and the PNPLA3 rs738409 and cluster of differentiation 14 (CD14) rs2569190 variants [23]. In the present study, although we did not include BMI in the predictive score, we managed to obtain good predictive accuracy for both sexes by replacing the CD14 variant with the HSD17B13 variant rs72613567.

## 5. Conclusions

Our new predictive score is important both from a prevention point of view and for future costing strategies. In fact, knowing the prevalence of the PNPLA3 rs738409 and HSD17B13 rs72613567 variants and the average age of onset of at-risk alcohol consumption in a certain population, it will be possible to deduce projections over time of the burden represented by cirrhotic patients for the healthcare system. Indeed, the cost of alcohol-related cirrhosis exceeds all other etiologies of cirrhosis [25]. Furthermore, the prognostic model can be used to motivate drinkers to stop drinking. Indeed, complete abstinence is the only effective therapeutic option for both compensated and decompensated cirrhosis and, when the disease progresses despite abstinence, liver transplantation is the only option but has very high costs [25,26].

The study has limitations due to the lack of external validation and its monocentric and retrospective nature. The latter aspect may have led to selection bias. However, prospective studies investigating the effects of alcohol consumption on the progression of liver disease would be too long and unfeasible. These studies would also not be ethically acceptable if we did not intervene by trying to interrupt the patients’ alcohol intake to observe the natural history of the disease. However, our diagnostic models, especially the one for compensated cirrhosis, even if inferred with a low number of cirrhotic patients, are useful in clinical practice. Our predictive score of future cirrhosis development is also useful in clinical practice, above all to stratify patients into subgroups in order to increase the targeted surveillance of the disease in subjects with genetic predisposition from a precision medicine perspective. Finally, our data are not generalizable to all heavy drinking patients with chronic liver disease and other coexisting etiologies of liver injury, except those with coexisting fatty liver disease associated with metabolic syndrome. In fact, we excluded from our study patients with coexisting etiologies such as, among others, viral, autoimmune, and cholestatic etiologies. In these patients, it is likely that participation in the disease progression of alcohol consumption is reduced and that of the genetic variants we considered is absent.

## Figures and Tables

**Figure 1 biomedicines-11-02132-f001:**
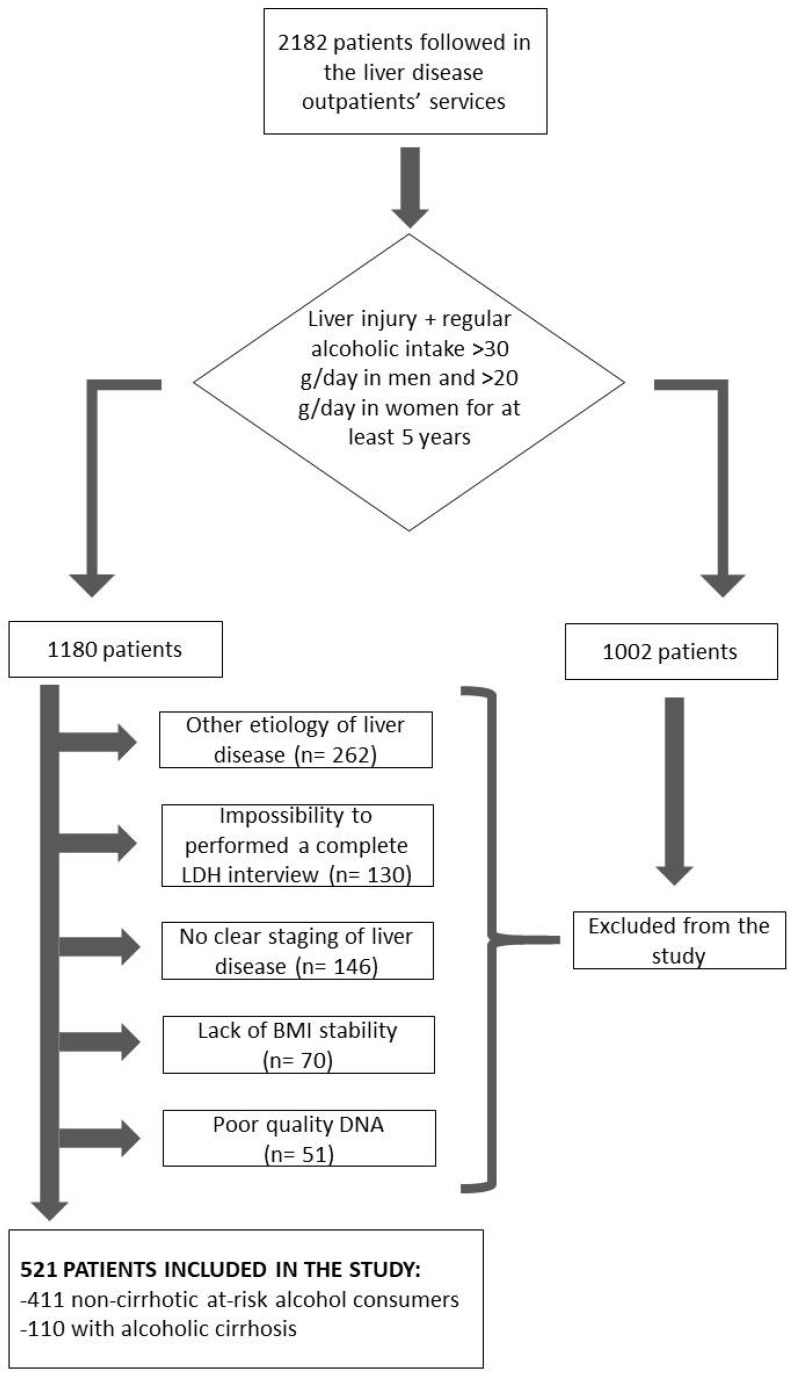
Study flow chart.

**Figure 2 biomedicines-11-02132-f002:**
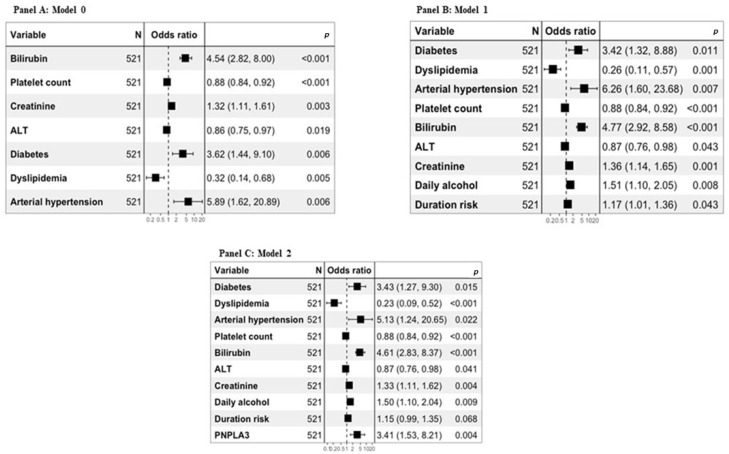
Multivariable logistic regression analysis for the entire study population. Panel A represents Model 0, panel B represents Model 1, and panel C represents Model 2. In panels (**A**–**C**), the models were defined by taking into consideration the values of platelet count (odds ratio expressed by counts × 103/µL divided by 10), creatinine (odds ratio expressed by mg/dL multiplied by 10), and ALT (odds ratio expressed by U/L divided by 10). In panels (**B**,**C**), the models were defined by taking into account the daily alcohol consumption (odds ratio expressed by the units of daily alcohol consumption divided by 10) and the duration of at-risk alcohol consumption (odds ratio expressed by the years of at-risk alcohol use divided by 5).

**Figure 3 biomedicines-11-02132-f003:**
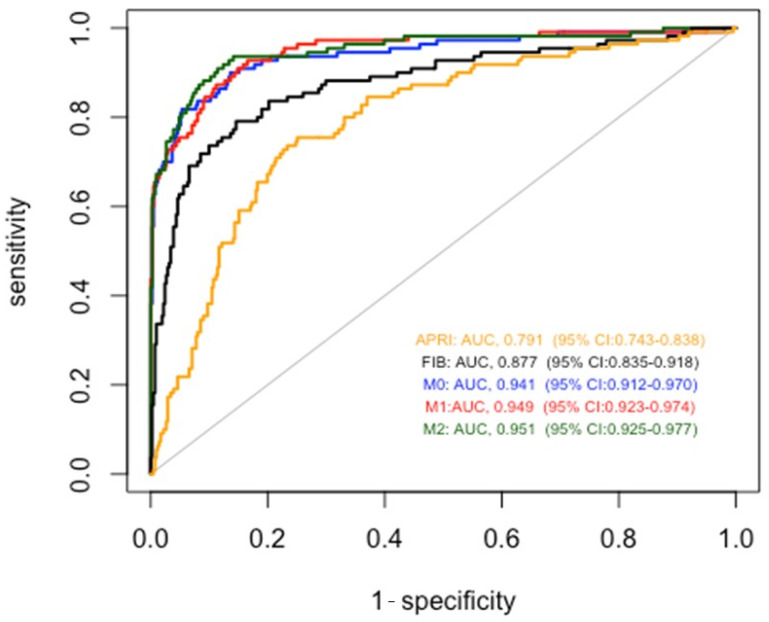
Comparison between accuracy of alcoholic cirrhosis diagnosis for the entire study population of models 0, 1, and 2 with respect to APRI and FIB4 scores.

**Figure 4 biomedicines-11-02132-f004:**
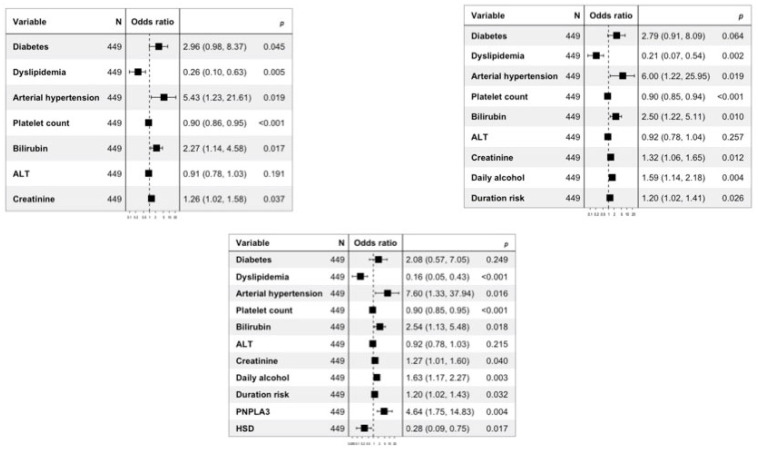
Multivariable logistic regression analysis for patients with compensated cirrhosis. Panel A represents Model 0, panel B represents Model 1, and panel C represents Model C. In panels, the models were defined taking into consideration the values of platelet count (odds ratio expressed by counts × 103/µL divided by 10), creatinine (odds ratio expressed by mg/dL multiplied by 10), and ALT (odds ratio expressed by U/L divided by 10). In panels, the models were defined by taking into account the daily alcohol consumption (odds ratio expressed by the units of daily alcohol consumption divided by 10) and the duration of at-risk alcohol consumption (odds ratio expressed by the years of at-risk alcohol use divided by 5).

**Figure 5 biomedicines-11-02132-f005:**
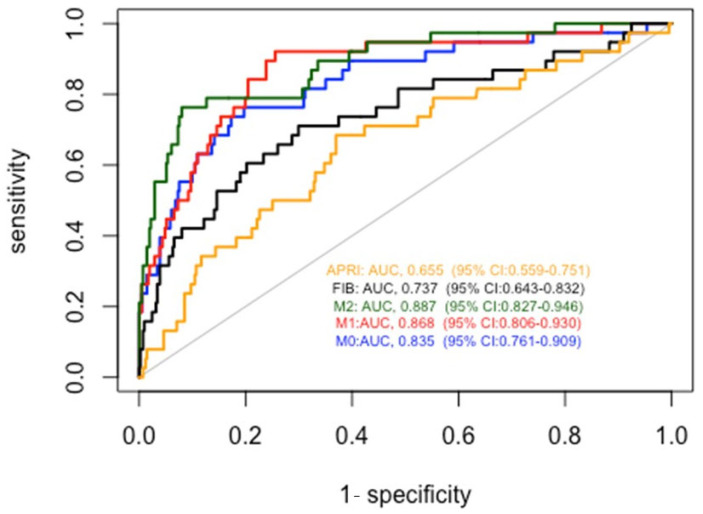
Comparison between accuracy of alcoholic cirrhosis diagnosis for patients with compensated cirrhosis of models 0, 1, and 2 with respect to APRI and FIB4 scores.

**Table 1 biomedicines-11-02132-t001:** Comparison between demographic and clinical data of non-cirrhotic patients with heavy alcohol consumption with respect to those of patients with alcoholic cirrhosis.

	All	Non Cirrhotic Subjects	All Cirrhotic Patients	Patients with Compensated Cirrhosis	*p*-Value (*)	*p*-Value (**)
		No	Yes			
N	521	411	110	38		
Age (y)	47.00 [40.00, 55.00]	45.00 [38.00, 53.00]	55.00 [47.25, 60.75]	52.50 [45.00, 60.50]	<0.001	0.001
Gender, male (%)	395 (75.8)	309 (75.2)	86 (78.2)	31 (81.6)	0.624	0.495
BMI	24.84 [22.40, 27.74]	24.60 [21.94, 27.30]	26.12 [23.88, 28.39]	25.41 [23.56, 27.36]	<0.001	0.184
Diabetes, n (%)	57 (10.9)	21 (5.1)	36 (32.7)	9 (23.7)	<0.001	<0.001
Obesity, n (%)	63 (12.1)	46 (11.2)	17 (15.5)	6 (15.8)	0.292	0.560
Age at onset of at-risk alcohol consumption (y)	22.00 [17.00, 29.00]	22.00 [17.00, 29.00]	21.00 [17.25, 27.75]	20.00 [17.25, 25.75]	0.875	0.309
Daily alcohol consumption (unit)	12.00 [8.00, 19.00]	13.00 [9.00, 20.00]	10.00 [7.94, 16.00]	11.00 [8.00, 17.50]	0.001	0.340
Duration of at-risk alcohol consumption (y)	21.00 [13.00, 30.00]	20.00 [12.00, 30.00]	29.50 [18.25, 40.00]	28.50 [20.00, 36.75]	<0.001	0.001
MELD score	NA	NA	12.13 [9.00, 17.42]	8.00 [6.43, 9.00]	NA	NA
INR	0.99 [0.93, 1.08]	0.98 [0.92, 1.02]	1.33 [1.13, 1.53]	1.06 [0.98, 1.19]	<0.001	<0.001
Serum bilirubin (mg/dL)	0.59 [0.41, 0.92]	0.53 [0.36, 0.72]	1.82 [0.90, 3.63]	0.88 [0.58, 1.27]	<0.001	<0.001
Serum creatinine (mg/dL)	0.80 [0.70, 0.90]	0.78 [0.70, 0.87]	0.88 [0.70, 1.00]	0.90 [0.70, 1.00]	0.0001	0.016
Platelet count × 10^3^/µl	227.00 [166.00, 283.00]	246.00 [198.00, 300.50.75]	85.50 [61.00, 147.25]	158.00 [82.50, 241.00]	<0.001	<0.001
Serum AST (U/L)	32.00 [21.00, 51.00]	29.00 [20.00, 50.00]	39.89 [28.00, 55.00]	33.50 [23.25, 50.75]	<0.001	0.440
Serum ALT (U/L)	28.00 [18.00, 46.00]	28.50 [18.00, 49.50]	25.00 [18.00, 39.00]	26.50 [17.25, 44.50]	0.100	0.363
Dyslipidemia	208 (39.9)	178 (43.3)	30 (27.3)	8 (21.1)	0.003	0.013
Hypertension	30 (5.8)	12 (2.9)	18 (16.4)	5 (13.2)	<0.001	0.007
PNPLA3, n (%)					<0.001	<0.001
II	211 (40.5)	191 (46.5)	20 (18.2)	5 (13.2)		
IM	235 (45.1)	177 (43.1)	58 (52.7)	20 (52.6)		
MM	75 (14.4)	43 (10.5)	32 (29.1)	13 (34.2)		
TM6SF2, n (%)					0.799	0.296
EE	445 (85.4)	350 (85.2)	95 (86.4)	32 (84.2)		
EK	73 (14.0)	59 (14.4)	14 (12.7)	5 (13.2)		
KK	3 (0.6)	2 (0.5)	1 (0.9)	1 (2.6)		
HSD17B13, n (%)					0.005	0.002
Wild type	333 (63.9)	251 (61.1)	82 (74.5)	32 (84.2)		
Heterozygous	165 (31.7)	144 (35.0)	21 (19.1)	3 (7.9)		
Mutant homozygous	23 (4.4)	16 (3.9)	7 (6.4)	3 (7.9)		

Legend: INR = International Normalized Ratio, MELD = Mayo end-stage liver disease; PNPLA3 = patatin-like phospholipase -3; TM6SF2 = transmembrane 6 superfamily member; HSD17B13 = hydroxysteroid 17-beta dehydrogenase 13. (*) Comparison between all cirrhotic patients and non-cirrhotic subjects. (**) Comparison between the compensated cirrhosis and no cirrhosis groups.

**Table 2 biomedicines-11-02132-t002:** Accuracy of the different predictors to diagnose alcoholic liver cirrhosis for the entire study population.

PREDICTOR	AUROC	95% CI	PREDICTOR vs. APRI	PREDICTOR vs. FIB4
APRI	0.791	0.743–0.838	-	<0.00001
FIB4	0.877	0.835–0.919	<0.00001	-
Model 0	0.941	0.912–0.970	<0.0001	0.000124
Model 1	0.949	0.924–0.974	<0.00001	<0.0001
Model 2	0.951	0.925–0.977	<0.00001	<0.0001

Legend: APRI = AST to Platelet Ratio Index; FIB 4 = Fibrosis-4 Index.

**Table 3 biomedicines-11-02132-t003:** Accuracy of the different predictors to diagnose alcoholic liver cirrhosis for patients with compensated cirrhosis.

PREDICTOR	AUROC	95% CI	PREDICTOR vs. APRI	PREDICTOR vs. FIB4
APRI	0.655	0.559–0.751	-	0.008965
FIB4	0.737	0.643–0.832	0.008965	-
Model 0	0.835	0.761–0.909	0.0005521	0.02265
Model 1	0.868	0.806–0.930	<0.00001	0.001347
Model 2	0.886	0.827–0.9461	<0.00001	0.000476

Legend: APRI = AST to Platelet Ratio Index; FIB 4 = Fibrosis-4 Index.

## Data Availability

Not applicable.

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
