# Peer review of "New Diagnostic and Prognostic Models for the Development of Alcoholic Cirrhosis Based on Genetic Predisposition and Alcohol History"

_biomedicines, 2023, doi:10.3390/biomedicines11082132_

Round 1
Reviewer 1 Report
This an outstanding study, well planned, well analysed and presented. My comments are for small improvements. The fact that the sample of interest is 38 is of concern, but the AUC curves are convincing. The text is in impeccable English.
Introduction
“Survival also depends 47 on the stage of the disease in which cirrhosis is diagnosed, with median values of 12 and 48 2 years respectively for subjects with compensated and decompensated cirrhosis [5]”
[Please describe the study design, numbers at commencement, attrition and causes of attrition, stage of disease and 95% confidence limits for survival to provide a complete picture to your readers].
Methods
“On the first visit, all patients 99 underwent a detailed clinical examination and interview on the clinical history and life- 100 style habits. At-risk alcohol consumption, defined as ≥3 and ≥2 alcohol units for men and 101 women respectively for at least 5 years, was present in 1180 patients. One unit of alcohol 102 was defined as 12 g of ethanol. In patients with at-risk alcohol use, a more thorough alco- 103 hol history was obtained by interview using the lifetime drinking history (LDH). In par- 104 ticular, the age reported at the beginning of the risky alcohol consumption, the duration 105 in years of the risky alcohol consumption and the average daily intake of alcohol ex- 106 pressed as the number of alcoholic units per day were obtained.”
[alcohol intake histories are notoriously minimised by some patients. Do you have alcohol histories taken at different times to compare reliability?]
[“≥3 and ≥2 alcohol units for men and 101 women respectively for at least 5 years” Is this daily intake?]
“we re- 124 peated the cross-sectional analysis limiting it, as far as cirrhotic patients are concerned, to 125 those with compensated cirrhosis.”
[How did you define compensated cirrhosis? What were the median values and IQRs for your decision?]
[How do the exclusions shown in your excellent flow chart affect the generalisability of your study? For the patients excluded how did their characteristics differ from those included?]
Results
“In par- 239 ticular, the AUROCs were 0.791 for APRI, 0.877 for FIB-4, 0.941 for Model 0, 0.949 for 240 Model 1 and 0.951 for Model 2. 241 When applied to the cross validation method considering 10 folds the AUROCs were 242 0.941 (95% CI: 0.913-0.968) for Model 0, 0.946 (95% CI: 0.921- 0.971) for Model 1 and 0.949 243 (95%CI: 0.923- 0.974) for Model 2.”
“The AUROC for the predictive 306 model of cirrhosis development at 24 years was 0.836 (95% CI: 0.755-0.918). The prognostic 307 score was defined as: (0.09*age at onset of at-risk alcohol consumption) + (0.86*number 308 of PNPLA3 variant alleles) -0.47* number of HSD17B13 variant alleles). The threshold of 309 3.34 performed in predicting the risk of alcoholic cirrhosis development, with a sensitivity 310 0.74 and a specificity of 0.75. The score was internally validated by bootstrap sampling 311 procedure, giving an AUC of 0.777 (95% CI: 0.702-0.845).”
[These are spectacular AUC results. Do you have results for other studies for comparison so you can compare your databases and duration of follow up?
[Can you make a picture that clinicians could use for counselling as an appendix to your article. Very visual with a person and perhaps many barrels of alcohol then a picture of a liver in different stages advancing to cirrhosis].
Author Response
REVIEWER 1
Introduction
R1: Please describe the study design, numbers at commencement, attrition and causes of attrition, stage of disease and 95% confidence limits for survival to provide a complete picture to your readers.
Authors: in accordance with this reviewer's request, we have better explained in the introduction the characteristics of the study (reference #5) which reported a median survival of 12 and 2 years respectively for subjects with compensated cirrhosis and for those with decompensated disease (lines 50-53 of the new version). The study is a systematic review of 118 studies. The review evaluated predictors of long-term survival (>6 months) in patients with cirrhosis. Of the studies analyzed in the review, 58% were prospective, 51% included consecutive patients, and 26% met the authors' criteria for a "good" quality study. No 95% confidence limits were reported for survival.
Methods
R1: alcohol intake histories are notoriously minimised by some patients. Do you have alcohol histories taken at different times to compare reliability?
Authors: we agree with this Reviewer that histories of alcohol use are notoriously minimised by some patients. Unfortunately we do not have alcohol histories taken at different times to compare reliability in the same patients. However, the co-authors of our study who collected lifetime drinking history (LDH) were experienced alcohol history specialists and therefore we are confident that the data collected is accurate. In cases where there was doubt that the patient minimized, the data on daily alcohol consumption was not entered and, therefore, the patient was excluded from the study because LDH was incomplete (see figure 1). In the new version we have specified the expertise of who collected the alcoholic history (lines 107-108 of the new version).
R1: “≥3 and ≥2 alcohol units for men and women respectively for at least 5 years” Is this daily intake?
Authors: we thank this Reviewer because we forgot to specify that we meant the units of alcohol consumed per day. In the new version we have added it (line 104 of the new version).
R1: How did you define compensated cirrhosis? What were the median values and IQRs for your decision?
Authors: the definition of compensated cirrhosis is well established and includes the past or present absence of ascites or variceal hemorrhage or hepatic encephalopathy or jaundice. In the new version we have added it (lines 130-131 of the new version).
R1: How do the exclusions shown in your excellent flow chart affect the generalisability of your study? For the patients excluded how did their characteristics differ from those included?
Authors: there were several reasons for exclusion from our study. Among these, the impossibility of collecting a complete LDH interview or of performing genotyping for poor quality DNA, the unclear staging of the liver disease and the lack of stability of the BMI would not have made it possible to investigate the endpoints of the study. As regards the possible generalizability of the results of the study to other populations, the patients that we excluded because they had other etiologies of liver damage other than alcohol and any components of the metabolic syndrome presented some similar and other different characteristics compared to the patients included in the study. Notably, in the excluded patients, the distribution of metabolic syndrome components between patients with and without cirrhosis was similar to that of the study population. In contrast, cirrhotic patients with other liver disease etiologies excluded from the study were younger than cirrhotic patients included in the study, started drinking later, and had fewer years of at-risk alcohol use. These differences accord with a lower relative importance of alcohol history for progression of liver disease when other causes of liver injury coexist. Consistent with this, among patients with other liver disease etiologies excluded from the study, there were no differences in genetic variants between cirrhotic and non-cirrhotic patients. However, we did not analyze these patients because their number was too small for the purposes of our study. Thus, regarding the generalizability of our data to all heavy drinking patients with chronic liver disease, our conclusions are not applicable to patients with other etiologies of liver injury except those with coexisting fatty liver disease associated with the metabolic syndrome. In fact, we excluded from our study patients with coexisting etiologies (i.e. viral, autoimmune, cholestatic) in which it is likely that the participation in the disease progression of alcohol consumption is reduced and that of the genetic variants we considered is absent. In the new version we have added these concepts (lines 394-400 of the new version).
Results
R1: These are spectacular AUC results. Do you have results for other studies for comparison so you can compare your databases and duration of follow up?
Authors: in a recent study, the AUCs for the diagnosis of cirrhosis of different genetic models with diabetes status ranged from 0.619 to 0.665 [reference # 16 of our study]. Our AUC values for the diagnosis of cirrhosis are higher for the greater number of clinical variables and the detailed alcoholic variables available. In the new version we have added these concepts (lines 345-349 of the new version).
R1: Can you make a picture that clinicians could use for counselling as an appendix to your article. Very visual with a person and perhaps many barrels of alcohol then a picture of a liver in different stages advancing to cirrhosis.
Authors: following this reviewer's suggestion, we have made a summary picture of the significance of our study (inserted as Supplementary figure 2)
Reviewer 2 Report
I have studied carefully the manuscript entitled "New diagnostic and prognostic models for the development of alcoholic cirrhosis based on genetic predisposition and alcohol history" by Mischitelli M. et al.
The manuscript deals with a topic of increased interest for the specialized reader. The manuscript is well-prepared and the language is fluent and accurate. The references cited are all relevant and up-to-date.
However, before considering acceptance for publication, the authors are wellcome to discuss mainly some methodological issues, as listed below:
Major issues
1. The data are reported as medians with interquartile ranges (IQR) for continuous variables (line 158). Moreover, the non-parametric Kruskal Wallis test was used for comparisons regarding baseline characteristics (line 159). Combined together, these options imply that all continuous variables of interest have a skewed distribution, else the use of mean and the parametric ANOVA test would have been prefered. The authors are requested to further elucidate this issue by i) using a proper normality test (e.g. Kolmogorov-Smirnov) for every continuous variable and ii) explaining why the major limitation of multivariable Logistic Regression used (line 169), namely the assumption of linearity between the dependent variable and the independent ones, is fulfilled, else choose a proper test (e.g. Generalized Linear Model).
Minor issues
1. In reference to data presented in Table 1, the authors are wellcome to discuss their approach towards potential multicollinearity issues.
2. Cross-sectional studies involve an one-time measurement of exposure and outcome, blurring any attempt to derive causal relationships. The authors are wellcome to clearly state this limitation in the relevant paragraph, ameding the phrase "The study has limitations due to its retrospective and monocentric nature" (line 365) for a more extensive and comprehensive one.
Author Response
REVIEWER 2
Major issues
- The data are reported as medians with interquartile ranges (IQR) for continuous variables (line 158). Moreover, the non-parametric Kruskal Wallis test was used for comparisons regarding baseline characteristics (line 159). Combined together, these options imply that all continuous variables of interest have a skewed distribution, else the use of mean and the parametric ANOVA test would have been preferred. The authors are requested to further elucidate this issue by i) using a proper normality test (e.g. Kolmogorov-Smirnov) for every continuous variable and ii) explaining why the major limitation of multivariable Logistic Regression used (line 169), namely the assumption of linearity between the dependent variable and the independent ones, is fulfilled, else choose a proper test (e.g. Generalized Linear Model).
R2: the authors are requested to further elucidate this issue by i) using a proper normality test (e.g. Kolmogorov-Smirnov) for every continuous variable
Authors: we thank this reviewer because in the statistical analysis section we had not reported, as we had practically done, that we had applied a suitable normality test (Shapiro Wilk test) to verify that the distribution of continuous variables was Gaussian. We verified that all continuous variables were not Gaussian and then described them by reporting the median and the interquartile range (IQR). In this context a nonparametric test is appropriate for comparison between groups. In the new version we have added these concepts (lines 165-168 of the new version).
R2: the authors are requested to further elucidate this issue by ii) explaining why the major limitation of multivariable Logistic Regression used (line 169), namely the assumption of linearity between the dependent variable and the independent ones, is fulfilled, else choose a proper test (e.g. Generalized Linear Model).
Authors: the dependent variable is a dichotomous and the independents variables are continuous or dichotomous, therefore it is not possible establish a linear relationship between dependent variable and independent variables. Considering the nature of dependent variable a generalized linear model was applied using as link function a logit function defined as the log(p/(1-p), where p is the probability of event (in this context the presence of cirrhosis). A generalized linear model with logit function is called a logistic regression model.
Minor issues
R2: in reference to data presented in Table 1, the authors are wellcome to discuss their approach towards potential multicollinearity issues.
Authors: in Table 1 there are the comparison between demographic and clinical data of non-cirrhotic patients with heavy alcohol consumption respect to those of patients with alcoholic cirrhosis, that is a univariable analysis. Therefore, each comparison is independent of each other and the multicollinearity cannot be a concern. The multicollinearity issue was considered when multivariable models were defined. In these models only the independent variables not highly correlated were considered. In the new version we have added the latter concept (lines 169-170 of the new version).
R2: Cross-sectional studies involve an one-time measurement of exposure and outcome, blurring any attempt to derive causal relationships. The authors are wellcome to clearly state this limitation in the relevant paragraph, ameding the phrase "The study has limitations due to its retrospective and monocentric nature" (line 365) for a more extensive and comprehensive one
Authors: we agree with this reviewer that in our study, as in any retrospective study, the strength of causal relationships is low and may suffer from selection bias. However, prospective studies investigating the effects of alcohol consumption on the progression of liver disease would be too long and unfeasible. These studies would also not be ethically acceptable if there were no interventions by trying to suspend the patients' alcohol intake in order to observe the natural history of the disease. In the new version we have added thes concepts (lines 385-389 of the new version).
Round 2
Reviewer 2 Report
I have carefully studied the revised manuscript entitled "New diagnostic and prognostic models for the development of alcoholic cirrhosis based on genetic predisposition and alcohol history" by Mischitelli M. et al.
The authors have performed a thorough revision and the quality of the manuscript has been certainly ameliorated. However, there are still some methodological and statistical issues that remain to be clarified.
Major issue
1) The logistic regression model is an example of a broad class of models known as generalized linear models. An assumption of logistic regression models is the absence of multicollinearity, or redundancy, among independent variables (see: Stoltzfus JC. Logistic regression: a brief primer. Acad Emerg Med. 2011;18:1099-1104. doi: 10.1111/j.1553-2712.2011.01185.x. PMID: 21996075). In case of all models presented in Figure 2, e.g. platelet count is highly correlated with bilirubin (see: Hancox SH, Smith BC. Liver disease as a cause of thrombocytopenia. QJM. 2013;106:425-431. doi: 10.1093/qjmed/hcs239. PMID: 23345462).
2) A limitation concerning the fact that the results have not been externally validated should be added in the relevant paragraph (line 378).
Minor issue
1. In a logistic regression model, a basic assumption that should be met is the linear relationship between any continuous independent variable and its logit-transformed outcome. This assumption is fulfilled when the interaction between each continuous independent variable and its natural logarithm is not statistically significant. The authors are wellcome to demonstrate that this assumption is met in the models they propose.
Author Response
Please see the attachment.
Kindly
F. Ferri
